# Impaired NRF2 Inhibits Recovery from Ischemic Reperfusion Injury in the Aging Kidney

**DOI:** 10.3390/antiox12071440

**Published:** 2023-07-18

**Authors:** Min Jee Jo, Ji Eun Kim, So Yon Bae, Eunjung Cho, Shin Young Ahn, Young Joo Kwon, Gang-Jee Ko

**Affiliations:** 1Department of Internal Medicine, Korea University College of Medicine, Korea University Guro Hospital, Seoul 08308, Republic of Korea; minjeeyoyo@naver.com (M.J.J.); beeswaxag@naver.com (J.E.K.); nepertary74@hanmail.net (S.Y.B.); icdej@naver.com (E.C.); sypooh712@naver.com (S.Y.A.); yjkwon@korea.ac.kr (Y.J.K.); 2Convergence Research Center for Development New Drug, Korea University College of Medicine, Seoul 08308, Republic of Korea

**Keywords:** aging, NRF2, ischemic reperfusion injury, recovery from acute kidney injury

## Abstract

Deteriorating kidney function is frequently observed in the elderly population, as well as vulnerability to acute kidney failure, such as ischemic/reperfusion injury (IRI), and inadequate recovery from IRI is one of the mechanisms of kidney dysfunction in the elderly. The potential mediators in the progression of kidney dysfunction in the aging kidney have not yet been clearly revealed. In this study, we investigated the role of nuclear factor erythroid 2-related factor 2 (NRF2), which is an essential regulator of cellular redox homeostasis, in restoring kidney function after IRI in the aging kidney. NRF2 expression decreased significantly in the kidneys of old mice, as well as histologic and functional renal recovery after IRI; 45-min renal pedicle clamping was retarded in old compared with young mice. Persistent renal injury during the recovery phase after IRI was aggravated in NRF2 knockout (KO) mice compared to wild-type mice. Oxidative stress occurred in NRF2 KO old mice during the IRI recovery phase along with decreased expression of mitochondrial OXPHOS-related proteins and a reduction in mitochondrial ATP content. In vitro, hypoxia/reoxygenation (H/R) injury was aggravated in senescent human proximal tubuloepithelial cells after NRF2 restriction using NRF2 siRNA, which also increased the level of oxidative stress and deteriorated mitochondrial dysfunction. Treating the mice with an NRF2 activator, CDDO-Me, alleviated the injury. These results suggest that NRF2 may be a therapeutic target for the aging kidney.

## 1. Introduction

As the human lifespan increases significantly worldwide, the number of people aged 65 years and older is estimated to exceed 800 million, and that trend is expected to accelerate in the future [1]. Decreased kidney function in the elderly is an important health issue, and kidney fibrosis, vascular insufficiency associated with atherosclerosis, and exposure to nephrotoxic medications and procedures have been suggested to contribute to the decline in kidney function with age.

Acute kidney injury (AKI) is characterized by rapid loss of kidney function within a few days or a week [2,3]. AKI is a serious health problem affecting up to 15.3% of all hospitalized patients [4]. Patients with AKI are at increased risk of chronic kidney disease (CKD), and a significant number of those patients progress to end-stage kidney disease [5,6]. AKI is prevalent and has been associated with high rates of morbidity and mortality, as well as medical expenses. The risk of mortality among hospitalized patients with AKI is three to six times higher than for individuals who do not suffer from AKI [7]. The elderly are more vulnerable to AKI due to the structural and functional decline of kidneys with age, and the prevalence of AKI appears to be higher in the elderly [8,9,10]. Moreover, the degree of AKI occurring in the elderly is often more severe and is associated with less chance of renal recovery, resulting in an increased risk of developing CKD [11]. One study showed that renal function did not recover in about 31% of patients with AKI aged over 65 years, which was a significantly higher rate than that in younger patients (26%) [12]. However, the mechanisms associated with the worsening of the severity of AKI or an increased chance of progression to CKD after AKI in the elderly remains unclear.

Nuclear factor E2-related factor2 (NRF2) is a key regulator of the cellular oxidative stress response. Coupled with Keap1, NRF2 induces the transcription of many antioxidant and cytoprotective genes [13,14]. Under stressful conditions, NRF2 detaches from Keap1, which rescues NRF2 from ubiquitination and enters the nucleus to trigger the transcription of target genes, including antioxidant and cytoprotective genes [15,16,17]. To protect kidney cells from stress, NRF2 plays an integrative role in driving the expression of genes encoding enzymes involved in producing antioxidants and reducing pro-oxidants [18,19]. The accumulation of reactive oxygen species (ROS) is an important mechanism of organ dysfunction with aging, and NRF2 plays a role in the aging kidney. In this study, we investigated the role of NRF2 during recovery from ischemic/reperfusion injury (IRI) in the aging kidney.

## 2. Materials and Methods

### 2.1. Animal Model

All experimental protocols and animal procedures were performed according to the National Institutes of Health (NIH) guidelines for the use of experimental animals, and this study was approved by the Korean University Institutional Animal Care and Use Committee (IACUC approval no: KOREA-2020-0040). Breeding pairs of Nrf2^+*/−*^ (B6.129 × 1-Nfe2 l2tm1Ywk/J) mice were purchased from Jackson Laboratory through Orient-Bio (Seongnam, Korea). Nrf2^−/−^ KO mice were generated after breeding, and genotyping was confirmed via polymerase chain reaction (PCR). Wild-type mice were purchased from Orient-Bio as the control group. Two-month-old mice were used as the younger group and twelve-month-old mice were used as the older group. AKI was induced using unilateral IRI by clamping unilateral renal blood vessels in mice for 45 min and then reperfusing the young WT (*n* = 8), old WT (*n* = 9), and old NRF2 KO mice (*n* = 9). The contralateral kidney was saved from IRI and used as a control. Recovery from renal injury was investigated 4 weeks after IRI. The NRF2 activator, CDDO-Me (3 mg/kg, #6646, Tocris, Bristol, UK), and vehicle (phosphate-buffered saline (PBS) with 10% DMSO and 10% Cremophor-EL) were administered by oral gavage every 3 days for 4 weeks from the third day of IRI. Increased expression of NRF2 was demonstrated in the CDDO-Me-treated group (*n* = 5) compared to the vehicle group (*n* = 5).

### 2.2. Measurement of Urinary Albumin Excretion

For the assessment of urinary albumin excretion, urine samples were collected in metabolic cages during a period of 24 h. Urinary creatinine was evaluated using a Creatinine Parameter Assay Kit (KGE005; R&D Systems Inc., Minneapolis, MN, USA) and urinary albumin was measured using a Mouse Albumin ELISA Kit (41-ALBMS-E01; ALPCO, Salem, NH, USA) according to the manufacturer’s instructions. The total urine albumin/creatinine ratio was used to calculate urinary albumin excretion.

### 2.3. Determining Oxidative Stress 

Lipid peroxidation is an index of oxidative stress and was assessed via malondialdehyde (MDA) production in kidney tissues using the thiobarbituric acid reactive substance test (OxiSelect MDA Adduct ELISA Kit; Cell Biolabs Inc., San Diego, CA, USA) according to the manufacturer’s instructions. 

### 2.4. Histological Examination and Immunohistochemical Analysis 

Formalin-fixed paraffin-embedded tissue blocks were sectioned to a thickness of 4 μm and stained with periodic acid-Schiff or Masson’s trichrome (M-T). The interstitial volume of the renal cortex and the degree of fibrosis were determined in sections stained with M-T. The kidney sections for immunohistochemical examination were hydrated in a graded ethanol series after deparaffinization and treatment with 0.1% trypsin (Zymed, San Francisco, CA, USA) and 0.3% H_2_O_2_, and incubated with blocking serum (Vector Laboratories, Peterborough, UK) to prevent nonspecific detection. The slides were incubated overnight at 4 °C with anti-TGF-β1 (1:100, Abcam, Cambridge, UK), followed by incubation with biotin-conjugated secondary antibodies. An avidin-biotin horseradish peroxidase complex and 3,3-diaminobenzidine substrate solution (Vector Laboratories, Burlingame, CA, USA) were applied to the slides at room temperature for colorization, and the slides were counterstained with hematoxylin (Sigma-Aldrich, St. Louis, MO, USA). Negative control slides were prepared by staining under identical conditions with rabbit serum as a substitute for the primary antibody. Interstitial collagen deposition was assessed via Sirius Red staining. After deparaffinization, the kidney sections were hydrated and incubated in Picrosirius Red solution (1% Sirius Red in saturated picric acid) for 18 h, followed by 0.01 N HCl treatment for 2 min and dehydration. All sections were examined in a blinded manner under a light microscope (Olympus BX-50; Olympus Optical, Tokyo, Japan). Finally, eight to ten high-power fields were captured (400× magnification), and M-T-, TGF-β1-, and Sirius Red-positively stained areas were determined using Image-Pro Plus version 5.3 (Media Cybernetics Inc., Silver Spring, MD, USA).

### 2.5. Isolation of Mitochondria

Mitochondria were isolated from mice kidney tissues using a mitochondria isolation kit (89801; Thermo Scientific, Rockford, IL, USA) according to the manufacturer’s instructions. Briefly, kidney tissues were minced with BSA Reagent A solution and homogenized. The homogenates were added to Mitochondria Isolation Reagent C and centrifuged at 700× *g* for 10 min. The supernatant was transferred to a new tube and centrifuged at 3000× *g* for 15 min. The pellet was the mitochondrial fraction, which was lysed with 2% CHAPS in Tris-buffered saline (TBS). The mitochondrial fraction was quantified and assessed using Western blot analysis.

### 2.6. Measurement of ATP Content

To assess the amount of mitochondrial ATP, ATP contents were measured using an ATP Assay Kit (ab83355; Abcam, Cambridge, UK) according to the manufacturer’s instructions following the isolation of mitochondria from kidney tissue. In brief, mitochondrial kidney pellets were homogenized and loaded in a 96-well-plate. ATP reaction mix was added to the sample and incubated at room temperature for 30 min in darkness. ATP contents were measured optical density with a microplate reader at 570 nm. 

### 2.7. Generation of Senescent Cells

Primary renal proximal tubule epithelial cells (RPTEC) derived from a healthy 41-year-old Asian male donor were purchased from the American Type Culture Collection (ATCC, Manassas, VA, USA). Long-term serial passaging of RPTECs was performed to generate senescent cells. The cells were seeded at a density of 5 × 10^5^ cells per dish and passaged every 5 days. Cell growth curves were obtained using the cumulative population doubling level (CPDL) value and by counting the number of cells. Cell counts were made with a LUNA-II TM automated cell counter (LUC-04-00507; Logos Biosystem Inc., Gyeonggi-do, Korea) every 5 days. The CPDL value was calculated based on the formula CPDL = X + 3.322 (logY − logI), where X is the initial population-doubling level, Y is the final cell yield, and I is the initial cell number seeded in the dish. 

### 2.8. Senescence-Associated β-Galactosidase (SA-β-gal) Staining

Cell senescence was verified with the Senescence β-Galactosidase Staining Kit (#9860; Cell Signaling Technology, Beverly, MA, USA) according to the manufacturer’s instructions. Briefly, the cells were seeded into a 6-well plate at a density of 2 × 10^5^ cells per well and incubated for 24 h. The cells were fixed with 1× fixative solution and incubated for 15 min at room temperature (RT). After rinsing in 1× PBS, the cells were stained with β-galactosidase staining solution and incubated overnight at 37 °C in an incubator without CO_2_. Cell images were captured using an optical microscope.

### 2.9. Cell Culture

RPTECs were grown and maintained in renal epithelial cell basal medium (PCS-400-030; ATCC) using the renal epithelial cell growth kit (PCS-400-040; ATCC) according to the manufacturer’s instructions. RPTECs for NRF2 KO were transfected with 0, 25, 50, and 100nM NRF2 siRNA (Silence^®^ Select siRNA, s9491, Invitrogen, Carlsbad, CA, USA) using Lipofectamine RNAiMAX reagent (Thermo Fisher Scientific, Waltham, MA, USA) for 24 h, and NRF2 expression was measured after siRNA treatment (Appendix A). 

### 2.10. Hypoxia/Reoxygenation Cell Model

Senescent RPTECs were seeded and incubated at 37 °C. Hypoxia and reperfusion injury was introduced to the cells under incubation with serum-free medium in a 1% O_2_ hypoxic chamber for 24 h followed by reoxygenation with fresh medium in a standard 5% CO_2_ incubator for 30 min. 

### 2.11. Western Blot Analysis

Cell lysates or tissues were lysed and quantified with a bicinchoninic acid (BCA) assay using the Pierce BCA Protein Assay Kit (#23225; Thermo Fisher Scientific). Sodium dodecyl sulfate-polyacrylamide gel electrophoresis was performed, and the proteins were transferred onto polyvinylidene difluoride membranes (Millipore, Milford, MA, USA). The membranes were blocked with 5% skim milk in TBS containing 0.1% Tween-20 for 1 h and incubated with primary antibodies against NRF2 (1:1000, Abcam), β-actin (1:10,000, Sigma), UCP2 (1:1000, Cell Signaling Technology), SDHA (1:1000, Cell Signaling Technology), RieskeFeS (1:1000, Santa Cruz Biotechnology, Dallas, TX, USA), COXI (1:1000, Santa Cruz Biotechnology), ATP5A (1:1000, Santa Cruz Biotechnology), HSP60 (1:1000, Cell Signaling Technology), and KIM-1 (1:1000, Novus Biologicals, Littleton, CO, USA) overnight at 4 °C. Then, the tissues were incubated for 1 h with horseradish-peroxidase-conjugated secondary antibody (PI-1000, PI-2000; 1:2000, Vector Laboratories), and the signal protein was detected using X-ray film and a chemiluminescence protocol. The blots were quantified and statistics were obtained using ImageJ (NIH, Bethesda, MD, USA).

### 2.12. Cell Viability Assay

Cell viability was evaluated with the Cell Titer 96 aqueous one-solution cell proliferation assay kit (G3580; Promega, Madison, WI, USA). RPTECs were seeded into a 96-well plate at a density of 1 × 10^4^ cells per well. The cells were stained with 20 μL MTS for 1 h. Cell viability was determined by measuring the OD at 490 nm using a microplate reader (Spectramax 190; Molecular Devices, Sunnyvale, CA, USA).

### 2.13. Measurement of Reactive Oxygen Species (ROS)

The cells were seeded into a white flat-bottom 96-well plate at a density of 1 × 10^4^ cells. The cells were stained with 10 μM carboxy-H2DCFDA (C400; Invitrogen) for 30 min at 37 °C. The stained cells were assessed via flow cytometry using FlowJo software (BD LSRFortessa™ X-20; BD Biosciences, San Diego, CA, USA). DCFDA fluorescence was evaluated at excitation/emission wavelengths of 495/515 nm using a fluorescence microplate reader (SpectraMax Gemini EM; Molecular Devices).

Cells used to measure mitochondrial ROS were seeded on a white flat-bottom 96-well plate at a density of 1 × 10^4^ cells. The cells were stained with 5 μM MitoSOX Red Mitochondrial Superoxide Indicator (M36008; Invitrogen) for 30 min at 37 °C. The stained cells were assessed via flow cytometry using FlowJo software (BD LSRFortessa™ X-20; BD Biosciences). Red fluorescence was evaluated at excitation/emission wavelengths of 510/580 nm using a fluorescence microplate reader. 

### 2.14. Assessment of the Mitochondrial Membrane Potential (MMP)

The cells were seeded in a white flat-bottom 96-well plate at a density of 1 × 10^4^ cells and stained with 10 μg/mL of JC-1 dye, which is an MMP probe (T3168, Invitrogen), for 30 min at 37 °C. The JC-1 monomer that formed was assessed at excitation/emission wavelengths of 514–529 nm and J-aggregate formation was evaluated by measuring excitation/emission wavelengths of 585/590 nm using a fluorescence microplate reader. 

### 2.15. Statistics

All experiments were conducted independently. Differences were examined with an unpaired *t*-test and two-way ANOVA test using GraphPad Prism 8 Software (La Jolla, CA, USA). A *p*-value of <0.05 was considered significant.

## 3. Results

### 3.1. NRF2 Expression Decreases in the Aging Kidney

NRF2 expression was decreased in the kidneys of older mice (12 months) compared to those of younger mice (2 months) (Figure 1A,B).

### 3.2. Delayed Renal Recovery after IRI in Old Mice

Renal recovery after IRI was examined 4 weeks after IRI via 45-min unilateral renal pedicle clamping in young and old mice. A histological examination with PAS staining showed mild expansion of the interstitium and atrophy of the tubules 4 weeks after IRI in the young mice, and the changes were accentuated in the old mice along with infiltration of inflammatory cells (Figure 2A). Tubular fibrosis detected using Masson’s trichrome (M-T) staining demonstrated the progression of tubular fibrosis in older mice compared to younger mice (Figure 2B). The degree of renal injury was also assessed via the albumin to creatinine ratio using 24-h urine collection with a metabolic cage. The urinary albumin to creatinine ratio (UACR) was increased at 4 weeks after IRI in old mice compared to the control; this difference was not demonstrated in young mice (young_con 19.6 ± 3.3 vs. young_IRI 15.7 ± 0.9, and old_con 23.0 ± 3.7 vs. old_IRI 63.6 ± 12.3 μg/g.cr, *p* < 0.05 compared to all other groups, Figure 2C). Changes in the level of malondialdehyde (MDA), as a marker of oxidative stress, were accompanied by tissue injury that increased 4 weeks after IRI compared to the control, and this was more prominent in older mice than younger mice (young_con 2.2 ± 0.3 vs. young_IRI 3.5 ± 0.2, *p* < 0.05 vs. young_con and old_con 3.2 ± 0.1 vs. old_IRI 4.6 ± 0.3 μm MDA/g/mL protein, *p* < 0.05 in old IRI compared to all other groups, Figure 2D). These results indicate that incomplete recovery from IRI was augmented in the aging kidney, and increased oxidative stress was implicated in the process. 

### 3.3. Exacerbation of Renal Dysfunction after IRI in NRF2 Knockout Mice

The role of NRF2 during renal recovery after IRI was examined in the old mice. As shown in Figure 3A, atrophy of tubules and infiltrating inflammatory cells with sclerotic changes in glomeruli were aggravated during the IRI recovery phase in NRF2 knockout (KO) mice compared to wild-type (WT) mice. These changes were accompanied by increased fibrosis as detected via M-T staining (old_WT_con 1.1 ± 0.6 vs. old_WT_IRI 5.5 ± 0.3% of total area, *p* < 0.05 compared to old_WT_con and old_KO_con 0.9 ± 0.6 vs. old_KO_IRI 12.5 ± 3.1% of total area, *p* < 0.01 in old_KO_IRI compared to all other groups, Figure 3B) and collagen deposition examined via Sirius Red staining (old_WT_con 0.5 ± 0.1 vs. old_WT_IRI 7.3 ± 1.3% of total area, *p* < 0.05 compared to old_WT_con and old_KO_con 0.9 ± 0.6 vs. old_KO_IRI 20.5 ± 9.7% of total area, *p* < 0.01 in old_KO_IRI compared to all other groups, Figure 3B). Transforming growth factor (TGF)-β1 was expressed in a similar pattern as the degrees of kidney injury and fibrosis in NRF2 KO mice compared to WT mice (old_WT_con 1.2 ± 0.6 vs. old_WT_IRI 7.2 ± 1.1% of the total area, *p* < 0.01 compared to old_WT_con and old_KO_con 0.3 ± 0.2 vs. old_KO_IRI 13.7 ± 1.9% of the total area, *p* < 0.01 in old_KO_IRI compared to all other groups, Figure 3B). Retarded functional recovery after IRI was also manifested in NRF2 KO mice as increased albuminuria after IRI compared to WT mice (old_WT_con 23.0 ± 3.7 vs. old_WT_IRI 63.6 ± 12.3, and old_KO_con 59.1 ± 19.1, vs. old_KO_IRI 175.5 ± 117.5 μg/g.cr, *p* < 0.05 compared to all other groups, Figure 3C). Oxidative stress increased during the IRI recovery phase of the older mice and was accentuated in NRF2 KO mice (old_WT_con 3.2 ± 0.1 vs. old_WT_IRI 4.6 ± 0.3 μm MDA/g/mL, *p* < 0.05 compared to old_WT_con and old_KO_con 3.6 ± 0.2 vs. old_KO_IRI 5.6 ± 0.4 μm MDA/g/mL protein, *p* < 0.05 in old_KO_IRI compared to old_WT_con and old_KO_con, Figure 3D). 

### 3.4. Reduced Expression of Mitochondrial Protein in NRF2 Knockout Mice

To evaluate mitochondrial functional processes and the synthetic response, the expression of several mitochondrial proteins was examined, including uncoupling protein-2 (UCP2) and mitochondrial electron transport complex proteins, which contribute to maintaining mitochondrial function. Expression of UCP2, succinate dehydrogenase complex flavoprotein subunit A (SDHA, complex II), RieskeFeS (complex III), cytochrome *c* oxidase subunit 1 (complex IV), and ATP synthase F1 subunit alpha (ATP5A, complex V) was assessed via immunoblotting analysis of kidney tissue (Figure 4A,B). The expression of the mitochondrial complex proteins involved in the oxidative phosphorylation (OXPHOS) system decreased significantly in the kidneys during recovery after IRI among NRF2 KO mice compared to WT mice. Furthermore, the amount of mitochondrial ATP was significantly reduced in old WT mice compared to old NRF2 KO mice with IRI (Figure 4C). These results demonstrate that NRF2 deficiency in aging kidneys is associated with disrupted mitochondrial OXPHOS and collapse of mitochondrial biogenesis, accompanied by a reduction in ATP.

### 3.5. NRF2 Deficiency Increases Oxidative Stress and Causes Mitochondrial Dysfunction in Senescent Cells

Senescent human renal proximal tubular epithelial cells (RPTECs) were generated to confirm the results of in vivo experiments. RPTECs were grown and counted every 5 days. The cumulative population-doubling level (CPDL) was calculated based on the formula presented in the Materials and Methods section. Cells were grown through long-term serial passaging for approximately 60 days. No further cell growth was detected after passage 7. Cells at passages 0 and 7 were compared using SA-β-gal staining to confirm that the cells at passage 7 were senescent (Appendix A). NRF2 expression was blocked using NRF2 small interfering RNA (siRNA), and transfection with 50nM siRNA was chosen to induce a significant reduction in NRF2 expression without significant cellular cytotoxicity (Appendix A). Cell viability was measured via MTS assay in senescent RPTECs. Cell viability decreased in senescent RPTECs after hypoxia/reoxygenation (H/R) injury was applied, and was accentuated under NRF2 deficiency following NRF2 siRNA treatment (Figure 5A). H/R injury in senescent RPTECs induced expression of the representative renal injury marker KIM-1, and NRF2 deficiency significantly augmented the expression of KIM-1 after H/R injury (Figure 5B). Cellular ROS levels increased significantly in senescent RPTECs after H/R injury as measured via carboxy-DCFDA dye staining, and a more significant increase in ROS was observed in NRF2-deficient cells (Figure 5C). As unbalanced ROS affect mitochondrial integrity, mitochondrial ROS were measured using the mitochondrial superoxide indicator MitoSOX. H/R injury enhanced mitochondrial ROS in senescent RPTECs, which increased further under NRF2 deficiency (Figure 5D). The membrane-permeant cationic carbocyanine dye JC-1 was used to detect changes in mitochondrial function in senescent H/R-injured RPTECs. The lower ratio of red-to-green JC-1 fluorescence indicated lower polarization of the mitochondrial membrane. A decrease in the red-to-green fluorescence intensity ratio was detected in senescent RPTECs with H/R injury; this ratio decreased more significantly under NRF2 deficiency (Figure 5E). 

### 3.6. CDDO-Me Treatment Contributes to Renal Recovery 

The role of NRF2 in renal recovery after IRI in old mice was examined using the NRF2 activator, CDDO-Me. Intraperitoneal and oral administration of CDDO-Me was compared with a two- or three-times weekly intraperitoneal injection and oral gavage. The highest and most consistent activation of NRF2 was manifested after oral gavage three times per week (Appendix A). Renal recovery after IRI was examined in 12-month-old mice with or without activation of NRF2 by administering the vehicle and CDDO-Me through oral gavage three times a week for 4 weeks from the third day of IRI. Tubular atrophy and inflammatory infiltration were alleviated in old mice treated with CDDO-Me during the IRI recovery phase compared to the vehicle-treated group (Figure 6A). Moreover, KIM-1 expression decreased in the kidneys of the CDDO-Me treated group (Figure 6B). To determine whether the protective effect of CDDO-Me on renal injury was related to mitochondrial function, the expression of mitochondrial proteins involved in OXPHOS was improved upon administration of CDDO-Me (Figure 6C,D). In addition, it was demonstrated that the amount of ATP generation in kidney mitochondria was enhanced in the CDDO-Me-administered group compared to the old IRI group (Figure 6E).

## 4. Discussion

In the present study, we revealed that NRF2 is crucial during the renal recovery of aging kidneys from AKI. The NRF2 expression level was lower in the kidneys of older mice than younger mice. The degree of tissue injury was worse in older mice during the recovery phase after IRI compared to younger mice, which manifested as tubular atrophy and fibrosis. NRF2 knockdown hindered renal recovery and accentuated tissue injury after IRI in old mice along with aggravation of oxidative stress and mitochondrial dysfunction. NRF2 deficiency also worsened the degree of hypoxia/reperfusion (H/R) injury in senescent cells. Increased ROS after H/R injury and oxidative stress after IRI was aggravated under NRF2 deficiency along with worsening mitochondrial dysfunction. Activating NRF2 during recovery after IRI mitigated kidney injury. 

The importance of aging-related healthcare is increasing as the aging population expands. A gradual decline in renal function occurs with advancing age due to the decrease in cortical mass with corresponding increases in glomerulosclerosis, interstitial fibrosis, tubular atrophy, and arteriosclerosis [20]. Additional structural changes, such as scarring and stiffness, emerge during aging and further impair kidney function. 

Moreover, the incidence and severity of AKI increase with age [21]. AKI is associated with an increased risk of mortality, hospitalization, and other comorbidities, such as cardiovascular events [22]. In addition to acute-phase problems associated with AKI, insufficient recovery from AKI, which results in a prolonged decrease in renal function and progression to CKD, is a contributing factor to the increase in the prevalence of reduced kidney function in the elderly [23]. Old age is a major risk factor for impaired renal recovery after AKI [12], and IRI is one of the most common and important causes of AKI in the elderly. Therefore, a better understanding of the recovery from IRI in the aging kidney is necessary to improve disease outcomes in the elderly.

NRF2 is suppressed during ubiquitination and proteasomal degradation by binding the Keap1-Cul3 complex; NRF2 triggers the production of genes involved in the antioxidant response when it is transported to and accumulates in the nucleus by binding antioxidant response elements to code cytoprotective antioxidant proteins and detoxifying enzymes. NRF2 is the master regulator of cellular redox homeostasis that protects cells from oxidative stress [24], which is the main cause of aging-related functional decline in organs. An unbalanced oxidant-to-antioxidant ratio occurs with aging, resulting in the accumulation of damaged macromolecules from oxidative stress [25,26,27]. Decreased NRF2 expression in aging mice, as demonstrated in our study, may play an important role in the deterioration of kidney function in the aging kidney due to aggravation of oxidative stress. In this study, aggravation of kidney injury and incomplete recovery after IRI were manifested in older mice compared to younger mice, and were shown as increased kidney fibrosis and collagen deposition, as well as increased oxidative stress accompanied by tissue injury in older mice. NRF2 deficiency led to the accentuation of oxidative stress and aggravation of tissue injury after IRI in old mice. Decreased NRF2 expression has been associated with aging in previous studies, although a discrepancy has been reported between tissue- and cell-specific locations [28,29]. NRF2 is believed to play a role in organ dysfunction in the elderly, such as in neurodegenerative diseases and cancer [26,30]. NRF2 deficiency has been reported in the aggravation of kidney injury [31,32]. Increased oxidative stress under NRF2 deficiency triggered an increase in inflammatory cytokines and inflammatory cell infiltration in an animal model of diabetic nephropathy, which resulted in prominent proteinuria and kidney fibrosis [31]. Decreased NRF2 expression was also manifested in chronic tubulointerstitial nephropathy of rats induced with adenine, resulting in augmented oxidative stress and increased levels of pro-inflammatory cytokines and chemokines [33]. The progression of interstitial fibrosis is a key histological feature of an aging kidney [20]. In the present study, increased interstitial fibrosis during the recovery phase after IRI was manifested in older mice compared to younger mice, and was accentuated under NRF2 deficiency. Previous reports examined the age-associated tissue injury of other organs, such as the heart and liver, and NRF2-deficiency-induced tissue fibrosis along with organ dysfunction [34,35]. The role of NRF2 was studied previously in aging kidneys using an NRF2 activator. Treatment with the NRF2 activator resveratrol attenuates kidney injury observed in old mice compared to young mice, as measured via proteinuria and fibrosis [36]. These findings suggest an important role of NRF2 in the aging kidney and offer the possibility of developing a new therapeutic strategy using NRF2 for renal aging. 

Our results showing that the activation of NRF2 by CDDO-Me in the old mice alleviated kidney damage, aided recovery after IRI, and reduced the expression of kidney injury markers after CDDO-Me treatment support the therapeutic possibility of using NRF2 in the aging kidney. NRF2 activators, such as nitroalkane [37], CDDO-Im [38], bardoxolone methyl (CDDO-Me or RTA-402) [39], omaveloxolone (RTA-408) [40], and sildenafil [41], have been studied in AKI models, particularly during the acute stage of IRI [42]. Although it has not been adopted as a new strategy for kidney disease due to safety issues associated with some incident cases of heart failure, the role of the NRF2 activator, CDDO-Me, in diabetic nephropathy was examined in a phase 3 trial, and new phase 2 and 3 trials using CDDO-Me are ongoing. Therefore, the possibility of using an NRF2 activator as a therapeutic agent in a real-world clinical environment is high, and further research is warranted to attenuate the injury in the aging kidney using activated NRF2 [43,44,45].

In the present study, mitochondrial dysfunction was implicated in incomplete recovery after IRI under NRF2 deficiency in the aging kidney. The expression of mitochondrial complex proteins involved in the OXPHOS system decreased during the recovery phase after IRI in kidneys from older NRF2 KO mice, compared to that in wild-type mice. This finding suggests that NRF2 deficiency in aging kidneys is associated with the disruption of mitochondrial OXPHOS and the collapse of mitochondrial biogenesis. These results were confirmed by an in vitro study using senescent human tubular cells and NRF2 siRNA treatment. Restricting NRF2 expression reduced tubular cell viability in senescent RPTECs. H/R injury significantly reduced cell viability and resulted in mitochondrial dysfunction manifested as a loss of mitochondrial membrane potential (MMP) and depolarization, which was aggravated after restricting NRF2 function. The importance of mitochondrial dysfunction in aging has been studied recently [46], particularly in association with NRF2 deficiency [26]. In a previous study, impaired mitophagy was documented in the aging kidney after IRI. Diminished MMP has been detected in mitochondria from the kidneys of old rats. Oxidative stress during aging resulted in the malfunctioning of renal mitochondria due to inappropriate autophagy [47,48], and tissue injury and mitochondrial dysfunction during aging were mitigated by activating NRF2 in a previous study [49]. Targeting oxidative stress and mitochondrial impairment has been used to prevent age-related kidney injury in several previous studies [50,51], and NRF2 may be the key factor by which to modulate the process in the aging kidney. 

Various factors have been investigated in association with high susceptibility to AKI and delayed recovery in the aging kidney. Decreased klotho [52,53,54] and peroxisome proliferator-activated receptor-γ (PPARγ) [55,56] levels and increased activation of Wnt [57,58] have been documented in the aging kidney. These changes may interact especially with NRF2; for example, NRF2 activation may partly compensate for aging kidney injury caused by a deficiency in α-klotho [32]. Cell cycle progression and oxidative stress are also important in the aging kidney [59]. Increased oxidative stress in aging has been correlated with an increase in systolic blood pressure and production of oxidant/antioxidant enzymes in the kidney [60]. Aging-associated kidney damage has been linked to upregulated endothelin A receptor, interleukin-6, and NADPH oxidase 2 expression as well as the generation of superoxide and downregulation of klotho, the ET B receptor, and manganese superoxide dismutase [61]. However, the exact mechanisms of kidney aging are not completely understood and further research in respect of potential contributors and prognostic factors is needed.

## 5. Conclusions

Overall, NRF2 expression decreased in aging kidneys and was implicated in the aggravation of kidney injury and incomplete recovery after IRI in old mice. NRF2 deficiency resulted in the exacerbation of oxidative stress and mitochondrial dysfunction, which contributed to renal fibrosis and collagen deposition. Activating NRF2 mitigated the kidney injury after IRI in old mice, which supports a possible therapeutic strategy for NRF2 activation in the aging kidney. Further research should be conducted in order to overcome problems associated with the aging kidney.

The English in this document has been checked by at least two professional editors, both native speakers of English. For a certificate, please see: http://www.textcheck.com/certificate/D1ODEd, (accessed on 20 February 2023).

## Figures and Tables

**Figure 1 antioxidants-12-01440-f001:**
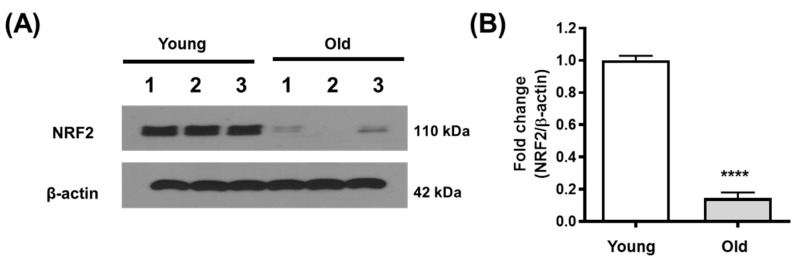
NRF2 expression decreased with age. Kidney tissues from 2-month-old mice as the younger group and 12-month-old mice as the older group were analyzed using Western blotting. β-actin was used as the loading control. Representative blots are shown on the left (**A**), and the densitometry of the blots is shown on the right (**B**). Error bars represent standard deviations. **** *p* < 0.0001 vs. young group. Abbreviations: young, young group; old, old group; NRF2, nuclear factor erythroid-2-related factor 2.

**Figure 2 antioxidants-12-01440-f002:**
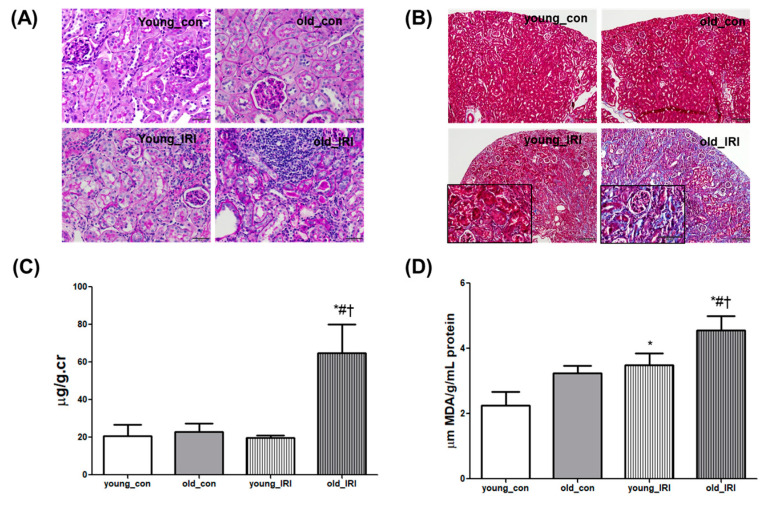
Exacerbation of renal injury and increased oxidative stress in the older IRI group. Representative histopathological images of kidney tissues from younger and older mice were examined using PAS (400× magnification, (**A**)) and M-T staining (100× magnification, 400× magnification in left lower box, (**B**)). (**C**) Urinary albumin/creatinine ratio in mice from young_con, old_con, young_IRI, and old_IRI groups. (**D**) MDA assay in kidney tissue from young_con, old_con, young_IRI, and old_IRI groups. * *p* < 0.05 vs. young_con group, # *p* < 0.05 vs. old_con group, † *p* < 0.05 vs. young_IRI group. Abbreviations: PAS, periodic acid-Schiff; M-T, Masson’s trichrome; IRI, ischemia-reperfusion injury; MDA, malondialdehyde.

**Figure 3 antioxidants-12-01440-f003:**
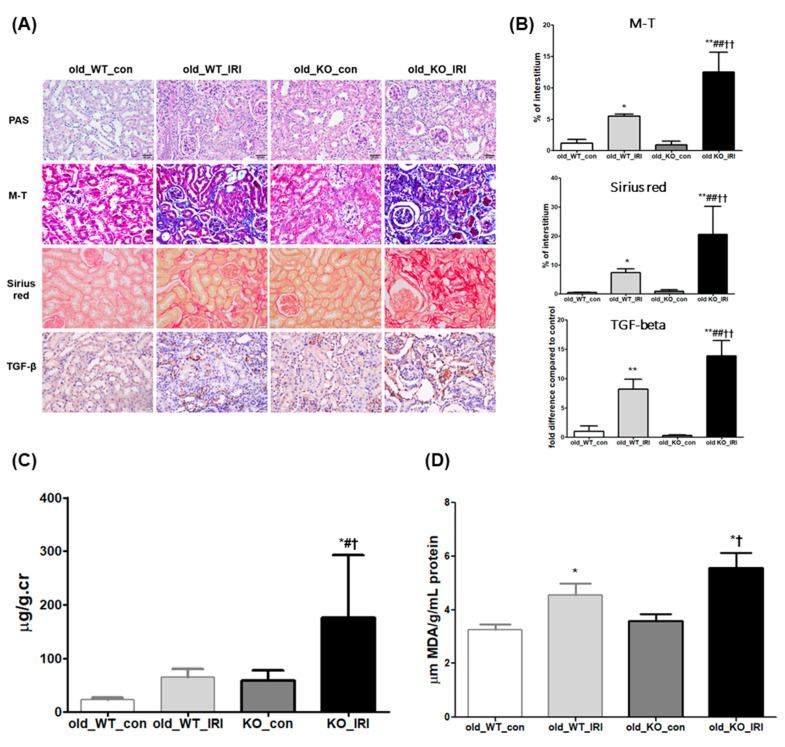
Renal histology in the recovery phase after IRI in wild-type (WT) and NRF2 knockout (KO) old mice. (**A**) Representative images of PAS-stained, M-T-stained, Sirius-Red-stained, and TGF-β1-stained kidney sections (400× magnification). (**B**) Quantitative analysis of the stained area. (**C**) Urine albumin/creatinine ratio in old_WT_con, old_WT_IRI, old_KO_con, and old_KO_IRI groups. Error bars represent standard deviation. (**D**) MDA assay of kidney tissues from old_WT and old_KO mice. Error bars represent standard deviations. The data of old_WT_con and old_WT_IRI used in Figure 3C,D were the same data used in Figure 2C,D. * *p* < 0.05, ** *p* < 0.01 vs. old_WT_con group, # *p* < 0.05, ## *p* < 0.01 vs. old_WT_IRI group, † *p* < 0.05, †† *p* < 0.01 vs. old_KO_con group. Abbreviations: PAS, periodic acid-Schiff; M-T, Masson’s trichrome; TGF-β1, transforming growth factor-β1; WT, wild type; IRI, ischemia-reperfusion injury; KO, knockout; MDA, malondialdehyde.

**Figure 4 antioxidants-12-01440-f004:**
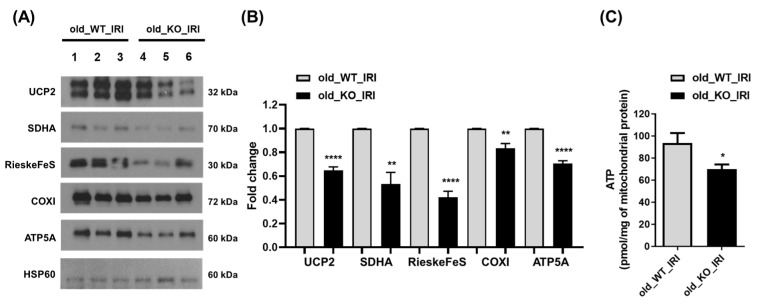
Mitochondrial proteins decreased in NRF2 KO mice during the recovery phase of IRI. Twelve-month-old WT and NRF2 KO mice underwent IRI to induce AKI. After 4 weeks of injury, mitochondria were extracted from kidney tissues using a mitochondria isolation kit. The mitochondrial fractions were assessed via Western blotting to determine the levels of mitochondrial proteins, including UCP2, SDHA, RieskeFeS, COXI, ATP5A, and HSP60. HSP60 was used as a mitochondrial loading control. Representative blots are shown on the left (**A**), and densitometric images of the blots are shown on the right (**B**). (**C**) Measurement of the ATP content was performed using an ATP assay kit to evaluate the amount of mitochondrial ATP in the kidney mitochondrial fraction. * *p* < 0.05, ** *p* < 0.01, **** *p* < 0.0001 vs. old_WT_IRI group. Abbreviations: IRI, ischemia-reperfusion injury; NRF2, nuclear factor erythroid-2-related factor 2; KO, knockout; AKI, acute kidney injury; UCP2, uncoupling protein-2; SDHA, succinate dehydrogenase complex flavoprotein subunit A; COXI, cytochrome c oxidase subunit 1; ATP5A, ATP synthase F1 subunit alpha; ATP, adenosine triphosphate.

**Figure 5 antioxidants-12-01440-f005:**
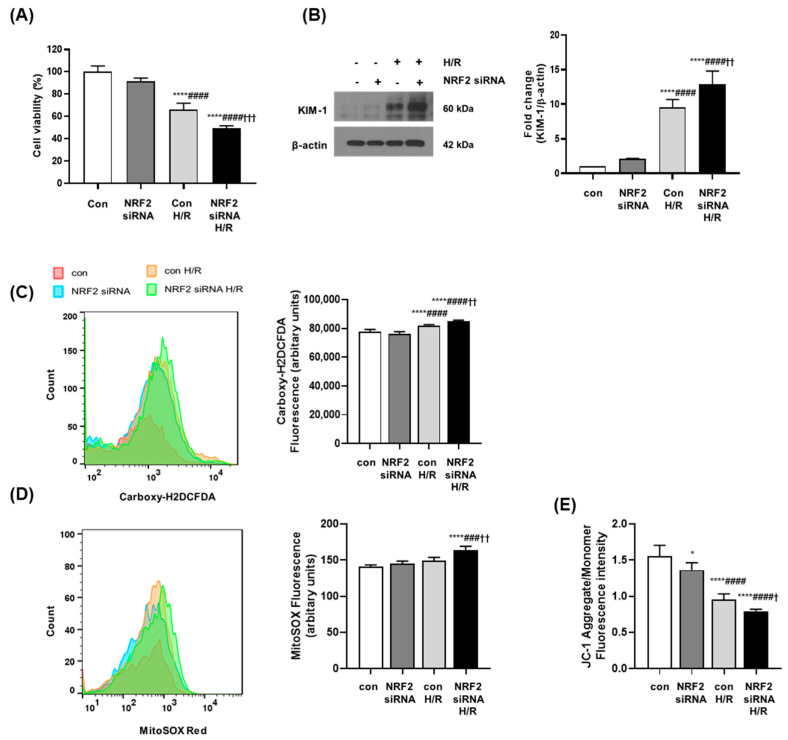
NRF2 deficiency decreased cell viability, increased KIM-1 expression and ROS production, and disrupted MMP in senescent H/R-injured RPTECs. (**A**) Senescent RPTECs were transfected with NRF2 siRNA, and H/R injury was induced in the transfected cells. Cell viability was measured using the cell titer 96 aqueous solution cell proliferation assay kit. Error bars represent standard deviations. **** *p* < 0.0001 vs. Con, #### *p* < 0.0001 vs. NRF2 siRNA, ††† *p* < 0.001 vs. Con H/R. (**B**) Cell lysates were assessed via Western blotting to investigate the expression of KIM-1. β-actin was used as the loading control. Densitometric analysis of the blots is shown for KIM-1. **** *p* < 0.0001 vs. Con, #### *p* < 0.0001 vs. NRF2 siRNA, †† *p* < 0.01 vs. Con H/R. (**C**,**D**) Cells were stained with carboxy-H2DCFDA dye or MitoSOX dye for 30 min in a 37 °C incubator. The stained cells were analyzed via flow cytometry and a fluorescence microplate reader. **** *p* < 0.0001 vs. Con, ### *p* < 0.001, #### *p* < 0.0001 vs. NRF2 siRNA, †† *p* < 0.01 vs. Con H/R. (**E**) The cells were incubated with JC-1 dye for 30 min at 37 °C. A fluorescence microplate reader was used to evaluate the MMP. Green fluorescence indicates the monomeric form and red fluorescence indicates the J aggregates. * *p* < 0.05 vs. Con, #### *p* < 0.0001 vs. NRF2 siRNA, † *p* < 0.05 vs. Con H/R. Abbreviations: NRF2, nuclear factor erythroid-2-related factor 2; KIM-1, kidney injury molecule-1; ROS, reactive oxygen species; MMP, mitochondrial membrane potential; RPTEC, primary renal proximal tubule epithelial cell; H/R, hypoxia/reoxygenation.

**Figure 6 antioxidants-12-01440-f006:**
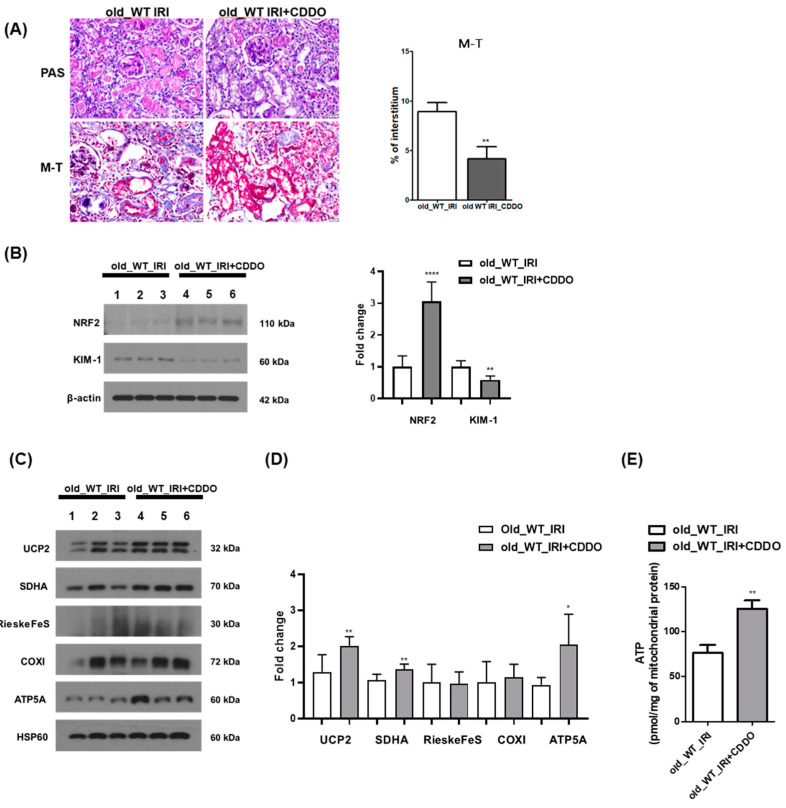
Effect of CDDO-Me treatment on recovery from IRI in old mice. (**A**) Histological analysis of old mice using PAS and M-T staining with a vehicle and CDDO-Me treatment (400× magnification). The 12-month-old mice were subjected to IRI injury. The mice were treated with 3 mg/kg of CDDO-Me via oral gavage three times a week for 4 weeks from the third day of IRI. Representative images are shown on the left, and the area of the interstitium is shown on the right. (**B**) Western blot analysis was performed to determine the expression of KIM-1 and NRF2. β-actin was used as the control. Error bars represent standard deviations. (**C**,**D**) Mitochondria were isolated from kidney tissues using a mitochondria isolation kit. The mitochondrial fractions were examined via Western blotting to determine the levels of mitochondrial proteins, including UCP2, SDHA, RieskeFeS, COXI, ATP5A, and HSP60. HSP60 was used as the mitochondrial loading control. Representative blots are shown on the left (**C**), and densitometric images of the blots are shown on the right (**D**). (**E**) Mitochondrial ATP in kidney tissue between Old_WT_IRI and Old_WT_IRI+CDDO group. * *p* < 0.05, ** *p* < 0.01, and **** *p* < 0.0001 vs. old_WT_IRI group. Abbreviations: CDDO-Me, bardoxolone methyl; PAS, periodic acid-Schiff; M-T, Masson’s trichrome; IRI, ischemia-reperfusion injury; AKI, acute kidney injury; KIM-1, kidney injury molecule-1; NRF2, nuclear factor erythroid-2-related factor 2; ATP, adenosine triphosphate.

## Data Availability

The data is contained within this article and Appendix A.

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
