# Peer review of "Impaired NRF2 Inhibits Recovery from Ischemic Reperfusion Injury in the Aging Kidney"

_antioxidants, 2023, doi:10.3390/antiox12071440_

Round 1
Reviewer 1 Report (Previous Reviewer 1)
The authors have addressed all comments raised by the reviewers successfully.
Author Response
We deeply appreciate you for your thorough review. The manuscript has been improved thanks to your precious comments.
Reviewer 2 Report (Previous Reviewer 2)
The authors addressed all my comments, performing additional experiments aimed at the evaluation of renal and mitochondrial function. The manuscript is improved. No further comments.
Author Response
We deeply appreciate you for your thorough review. The manuscript has been improved thanks to your precious comments.This manuscript is a resubmission of an earlier submission. The following is a list of the peer review reports and author responses from that submission.
Round 1
Reviewer 1 Report
The present study is an interesting study demonstrating the effect of NRF2 on IRI in the aging kidney. The following comments should be addressed:
1) Authors are showing a huge difference in NRF2 expression between young and old mice, almost complete absence, in the latter. Did they observe any other effects on the phenotype such as proteinuria, or effects on histology before the IRI model?
2) The same applies for the Nrf2-/- mice used. Authors need to assess their phenotype in terms of histological alterations and proteinuria prior to applying the IRI model.
3) Figure 4 western blotting. What was used for loading control? If HSP60 was used authors need to specify this in the legend.
4) Use of siRNA for Nrf2 in RPTECs resulted in an almost 50% reduction of protein expression. Did authors try other concentrations of siRNA to improve silencing?
5) Did authors assess possible effects via other Nrf2 regulated genes such as that of HO-1? Upregulation of HO-1 has been shown to be protective in IRI in mice Chen et.al doi: 10.1016/j.bbadis.2015.07.018. A potential knock on effect of Nrf2 absence should also be investigated.
Reviewer 2 Report
The authors demonstrated that NRF2 expression decreases in the kidney of old mice. Old mice undergoing IRI show an increased renal injury that further increases in knockout mice for NRF2 (NRF2 KO). Mitochondrial proteins decrease in NR2 KO mice undergoing IRI and mitochondrial dysfunction is observed in renal cells undergoing hypoxia/reoxygenation in vitro Finally, NRF2 activation through CDDO-Me rescues renal injury in NRF2 KO mice in response to IRI.
The work is interesting and suggests that NRF2 represents a potential therapeutic target for preventing renal aging. Results and the experimental design is well organized. I have some suggestions, aimed at improving the quality of the manuscript:
Major
1) In addition to morphological analyses, I suggest evaluating in mice functional parameters of renal function, such as proteinuria level, glomerular filtration rate, renal clearence
2) The evaluation of mitochondrial function in vivo is not sufficient and should be supported by the evaluation of ATP content, ETC activity.
3) The authors should demonstrate that the protective effects of CDDO-Me are also mediated by the reduction of mitochondrial dysfunction
